# Orthohantavirus Isolated in Reservoir Host Cells Displays Minimal Genetic Changes and Retains Wild-Type Infection Properties

**DOI:** 10.3390/v12040457

**Published:** 2020-04-17

**Authors:** Tomas Strandin, Teemu Smura, Paula Ahola, Kirsi Aaltonen, Tarja Sironen, Jussi Hepojoki, Isabella Eckerle, Rainer G. Ulrich, Olli Vapalahti, Anja Kipar, Kristian M. Forbes

**Affiliations:** 1Zoonosis Unit, Department of Virology, Medicum, University of Helsinki, 00290 Helsinki, Finland; teemu.smura@helsinki.fi (T.S.); paula.ahola@helsinki.fi (P.A.); kirsi.aaltonen@helsinki.fi (K.A.); Tarja.Sironen@helsinki.fi (T.S.); jussi.hepojoki@helsinki.fi (J.H.); olli.vapalahti@helsinki.fi (O.V.); 2Department of Basic Veterinary Sciences, Faculty of Veterinary Medicine, University of Helsinki, 00790 Helsinki, Finland; anja.kipar@uzh.ch; 3Laboratory for Animal Model Pathology, Institute of Veterinary Pathology, Vetsuisse Faculty, University of Zurich, 8057 Zurich, Switzerland; 4Institute of Virology, University of Bonn Medical Centre, 53012 Bonn, Germany; isabella.eckerle@chuge.ch; 5Institute of Novel and Emerging Infectious Diseases, Friedrich-Loeffler-Institut, Federal Research Institute for Animal Health, 17493 Greifswald-Insel Riems, Germany; rainer.ulrich@fli.de; 6Department of Biological Sciences, University of Arkansas, Fayetteville, NC 72701, USA; kmforbes@uark.edu

**Keywords:** immunity, Puumala orthohantavirus, rodent reservoir, spillover, vole, zoonoses

## Abstract

Orthohantaviruses are globally emerging zoonotic pathogens. While the reservoir host role of several rodent species is well-established, detailed research on the mechanisms of host-othohantavirus interactions has been constrained by the lack of an experimental system that is able to effectively replicate natural infections in controlled settings. Here we report the isolation, and genetic and phenotypic characterization of a novel Puumala orthohantavirus (PUUV) in cells derived from its reservoir host, the bank vole. The isolation process resulted in cell culture infection that evaded antiviral responses, persisted cell passaging, and had minor viral genome alterations. Critically, experimental infections of bank voles with the new isolate resembled natural infections in terms of viral load and host cell distribution. When compared to an attenuated Vero E6 cell-adapted PUUV Kazan strain, the novel isolate demonstrated delayed virus-specific humoral responses. A lack of virus-specific antibodies was also observed during experimental infections with wild-type PUUV, suggesting that delayed seroconversion could be a general phenomenon during orthohantavirus infection in reservoir hosts. Our results demonstrate that orthohantavirus isolation on cells derived from a vole reservoir host retains wild-type infection properties and should be considered the method of choice for experimental infection models to replicate natural processes.

## 1. Introduction

Understanding how zoonotic pathogens are maintained and transmitted in nature is critical for efforts to combat human disease [1]. Most investigations have focused on the identification of wildlife reservoir hosts [2,3], and to a lesser extent, the characterization of high-risk subgroups and time periods, such as migration and breeding seasons [4,5], when transmission is greatest in reservoir host populations and the risk of human exposure may therefore be elevated. However, it remains largely unknown how zoonotic pathogens are maintained at an individual-level in wildlife reservoirs, such as the molecular and cellular mechanisms of host-pathogen interactions that promote clearance or persistence, and feed back onto these population-level dynamics.

Hantaviruses (family *Hantaviridae*, order *Bunyavirales*) provide one of the most clearly described reservoir host-zoonotic pathogen relationships [6]. Hantavirus species have been identified across the globe, some of which cause Hemorrhagic Fever with Renal Syndrome (HFRS) in Eurasia or Hantavirus Cardiopulmonary Syndrome (HCPS) in the Americas [7]. In contrast to human infection, hantavirus infections are thought to be persistent and asymptomatic in their wildlife reservoirs [6]. Each hantavirus species is carried by a specific reservoir host; for example, Puumala orthohantavirus (PUUV), which is present in Europe and causes thousands of human infections annually [8], is carried by the bank vole (*Myodes glareolus*).

A major issue hindering detailed research on the molecular mechanisms of hantavirus infections in animal models has been the absence of a standardized experimental infection system [9]. Two main experimental strategies have been employed by researchers to date: (1) lung homogenates from naturally infected rodent hosts have been used as virus inoculum. However, this method suffers from slowly developing infections and a lack of standardized infection loads [10,11], and (2) hantaviruses have been isolated and grown to high titers in cell culture. However, this method has so far been performed almost exclusively using Vero E6 cells, which generates cell culture-adapted viruses with attenuated infection properties [12,13,14,15].

The purpose of this study was to develop an effective orthohantavirus experimental wildlife system to replicate natural infection processes. To these means, we employed PUUV and its reservoir host, the bank vole. Our objectives were to isolate PUUV from wild bank voles using a host-derived cell line, characterize the cell culture infection in terms of virus persistence and induced antiviral responses, and to evaluate genotypic changes of the virus during the isolation process using next-generation sequencing (NGS). Experimental bank vole infections were then used to compare the phenotype of the novel isolate to the wild PUUV–containing lung homogenate (PUUV-wt), an attenuated Vero E6 cell-adapted PUUV-Kazan strain and to PUUV in naturally infected wild voles using virus-specific RT-PCR, pathology, immunohistochemistry and serology.

## 2. Materials and Methods

### 2.1. Establishment of the Permanent Mygla.REC.B Cell Line

A live-trapped bank vole was anaesthetized with isoflurane and euthanized using CO_2_. After dissection, the kidney was washed in sterile PBS and minced with a blade. Tissue was resolved with 37 °C Dulbecco’s minimum essential medium-high glucose (DMEM; Sigma-Aldrich, St. Louis, MO, USA), supplemented with 10% inactivated fetal calf serum (FCS), 100 IU/mL Penicillin, 100 µg/mL Streptomycin, 2 mM L-glutamine, a mix of non-essential amino acids (Sigma-Aldrich, Sigma-Aldrich, St. Louis, MO, USA) and 1% Ofloxain (Tarivid; Sigma-Aldrich, St. Louis, MO, USA), and seeded into 6-well plates. After 5–7 days, primary cell growth occurred and the medium was replaced. Primary cells were immortalized when confluency of 40%–50% was reached using a lentiviral system carrying the large T antigen of SV40, as described previously [16]. Once an increase in cell proliferation was observed, cells were further passaged and preserved for long-term storage via cryo-freezing.

### 2.2. Virus Isolation and Propagation

For virus isolation, a serologically confirmed PUUV-infected bank vole from Suonenjoki, Finland was euthanized via cervical dislocation and lung samples were collected and frozen at −70 °C. The frozen lungs were then homogenized with a mortar and pestle in 1 mL of phosphate-buffered saline (PBS) over dry ice. After thawing, 500 µL of the homogenate was incubated for 1 h with semi-confluent Mygla.REC.B cells, which were grown as described above. Mygla.REC.B cells were passaged at 3-day intervals until 100% of cells were found to be infected with PUUV via an immunofluorescence assay as described below. We named the newly isolated hantavirus PUUV-Suonenjoki (PUUV-Suo) due to the geographical location of the bank vole trapping site. Note that although separate isolations were attempted from the lungs of several seropositive bank voles, successful virus isolation only occurred in one case.

An attenuated PUUV isolate, PUUV-Kazan strain [17], which was previously isolated and propagated in Vero E6 cells (green monkey kidney epithelial cell line; ATCC no. CRL-1586), served as a comparison for experimental infections. PUUV-Kazan was grown in Minimum Essential Medium (MEM; Sigma-Aldrich, Sigma-Aldrich, St. Louis, MO, USA) and supplemented with 10% inactivated FCS, 100 IU/mL of Penicillin and 100 µg/mL of Streptomycin and 2 mM of L-glutamine. Both PUUV-Suo and PUUV-Kazan strains were purified from supernatants of infected cells through a 30% sucrose cushion by ultracentrifugation and resuspended in corresponding growth medium.

Virus titers were measured by incubating diluted virus stocks with Vero E6 cells for 24 h at 37 °C, followed by acetone fixation and staining with a rabbit polyclonal antibody specific for PUUV nucleocapsid (N) protein (anti-PUUN; see below) and AlexaFluor488-conjugated donkey anti-rabbit secondary antibody (Thermo Scientific, Waltham, MA, USA). Fluorescent focus-forming units (FFFU)/mL were counted under a UV microscope (Zeiss Axio Imager 1; Zeiss, Jena, Germany). For a separate control inoculum, UV inactivation of PUUV-Suo was carried out using a Stratalinker UV crosslinker (Stratagene, San Diego, CA, USA) at 300,000 μJ/cm^2^.

### 2.3. Next-Generation Sequencing

Prior to RNA extraction, the bank vole lung homogenate and Mygla.REC.B cell culture (passage 20 after initial inoculation with wild virus) supernatant were treated with a cocktail of micrococcal nuclease (New England BioLabs, Ipswich, MA, USA) and benzonase (Millipore, Burlington, MA, USA) for 1 h at 37°C. RNA was extracted with Trizol and ribosomal RNA was removed using a NEBNext rRNA depletion kit (New England BioLabsaccording to manufacturer instructions. The sequencing library was prepared using a NEBNext Ultra II RNA library prep kit (New England BioLabs) and quantified using a NEBNext Library Quant kit for Illumina (New England BioLabs. Pooled libraries were then sequenced on a MiSeq platform (Illumina, San Diego, CA, USA) using a MiSeq v2 reagent kit with 150 bp paired-end reads. Raw sequence reads were trimmed and low-quality (quality score <30) and short (<50 nt) sequences were removed using Trimmomatic [17]. Thereafter, de novo assembly was conducted using MegaHit, followed by re-assembly against the de-novo assembled consensus sequences using BWA-MEM implemented in SAMTools version 1.8 [18]. The newly identified PUUV-Suo genomic sequences derived from bank vole lungs prior to virus isolation and from persistently infected Mygla.REC.B supernatant were uploaded to GenBank with the following accession numbers: MT024590 S segment bank vole lung, MT024591 M segment bank vole lung, MT024592 L segment bank vole lungs, MT024593 S segment cell culture supernatant, MT024594 M segment cell culture supernatant and MT024595 L segment cell culture supernatant.

### 2.4. Phylogenetic Analysis

The complete L-, M- and S-segment sequences were downloaded from GenBank and aligned using the MUSCLE program package [19]. A substitution model was estimated using jModeltest2 [20]. Phylogenetic trees were then constructed using the Bayesian Markov chain Monte Carlo (MCMC) method, implemented in Mr Bayes version 3.2 [21] using a GTR-G-I model of substitution with 2 independent runs and four chains per run. The analysis was run for 500,000 states and sampled every 5000 steps.

### 2.5. Detection of Infected Cells in Cell Culture

PUUV-infected cells, grown on a black 96-well plate (Nunc), were detected by an immunofluorescence assay after fixation with 4% paraformaldehyde and subsequent blocking/permeabilization with 3% BSA and 0.1% Triton X-100 (both steps 15 min). For this a polyclonal rabbit anti-PUUN serum was used, diluted 1:1000 in PBS, followed by incubation with AlexaFluor488-conjugated donkey anti-rabbit secondary antibody. Nuclei were stained using Hoechst 33420 diluted in PBS. Images were taken using a Perkin Elmer Opera Phenix confocal microscope (Finnish Institute of Molecular Medicine, University of Helsinki).

### 2.6. Detection of Antiviral Responses in Cell Culture

Non- and PUUV-Suo infected Mygla.REC.B cells were exogenously stimulated with TLR-3 ligand polyI:C (10 µg/mL, Sigma-Aldrich) or Sendai virus (multiplicity of infection 1; received from Prof. Ilkka Julkunen; National Institute of Health and Welfare, Helsinki, Finland). Cells were collected 1-day post-treatment, and RNA was extracted from cell cultures with Trisure (Bioline, London, UK) following the manufacturers’ instructions. Extracted RNA was then reverse transcribed to complementary DNA (cDNA) using random hexamers and RevertAid reverse transciptase (Thermo Scientific). Relative quantitative PCR was performed with Maxima SYBR Green master mix (Thermo Scientific) using AriaMx instrumentation (Agilent, Palo Alto, CA, USA). Published primer sequences were used to measure the mRNAs levels of the interferon-inducible gene were Mx2, a marker of innate immune activation towards virus infection [22], and actin, which is commonly used for normalization of mRNA levels between samples [23]. Fold changes of individual mRNA expression levels relative to mock-infected controls were performed by the comparative CT method [24].

### 2.7. Protein Content Analysis in Virus Preparations

The sucrose-cushion purified viruses PUUV-Suo and PUUV-Kazan were run on 4%–20% Sodium dodecyl sulfate-polyacrylamide (SDS-PAGE) gel (Bio-Rad, Hercules, CA, USA) and either stained with Coomassie-based Page Blue protein staining solution (Thermo Scientific) or transferred onto nitrocellulose membrane. The membrane was probed with polyclonal rabbit antiserums against one of Gn, Gc or N protein [25], followed by IRdye800-labelled donkey anti-rabbit secondary antibody (Li-Cor, Lincoln, NE, USA) in Tris-buffered saline (50 mM Tris pH 7.5; 150 mM NaCl, 0.005% Tween-20) containing 1.5% milk powder. The membranes were analyzed by an Odyssey imaging system (Li-Cor). 

### 2.8. Experimental Infections

Wild-captured voles from a PUUV endemic region in Central Finland were transported to the University of Helsinki BSL-3 facility and acclimatized to individually ventilated biocontainment cages (ISOcage; Scanbur, Karlslunde, Denmark) for two days with ad libitum water and food (rodent pellets and small pieces of turnip every second day). Prior to treatments, a small blood sample was collected from the retro-orbital sinus of each animal, and their PUUV infection status was determined by immunofluorescence assay (see below).

PUUV seronegative voles were divided into five treatment groups and subcutaneously injected with: (1) 100 µL of PBS (mock infection; *n* = 2 per time point), (2) 10,000 FFFUs of PUUV-Suo (*n* = 4 per time point), (3) UV-inactivated PUUV-Suo (*n* = 2 per time point), (4) 10,000 FFFUs of PUUV-Kazan (*n* = 2–3 per time point), or (5) a pooled homogenate prepared from the lungs of five PUUV-seropositive wild bank voles (PUUV-wt; *n* = 4 per time point). A total of 43 individual voles were used across the different time points and experimental treatments. 

The lung homogenate was obtained after grinding frozen lung tissue from naturally PUUV-infected voles (confirmed by PUUV-specific RT-PCR) with mortar and pestle in 1 mL of PBS on dry ice. Blood samples from all treatment groups were collected from the retro-orbital sinus at 1-week (wk) intervals post-infection (pi). Urine samples were collected from PUUV-Suo infected voles at 3d, and 1 and 2 wks pi. Voles were sacrificed using Isoflurane anesthesia, followed by cervical dislocation at 3d pi and 1 (7d), 2 (14–16d), and 5 wks (35–38d) pi to collect samples for viral RNA load and distribution analyses, pathology, immunohistology and gene expression assays.

### 2.9. Virus Quantification

Following euthanasia and dissections, RNA extractions of bank vole tissues and urine were performed using Trisure (Bioline) according to the manufacturers’ instructions, with 10 µg/mL glycogen as carrier. RNA was directly subjected to PUUV S-segment RT-qPCR analysis based on a previously described protocol [26], with TaqMan fast virus 1-step master mix (Thermo Scientific) using AriaMx instrumentation (Agilent).

### 2.10. Histological and Immunohistological Examinations

Two wild-trapped, naturally PUUV-infected adult bank voles were dissected and samples from the brain, heart, lung, liver, kidneys and spleen were fixed in 10% neutral-buffered formalin. Similarly, lung, liver, spleen and kidney samples were collected and formalin-fixed from each two PUUV-Suo infected voles euthanized at 3 d, 1 wk, 2 wks, and 5 wks pi and two PUUV-wt infected bank voles euthanized at 5 wks pi. The latter two voles were initially frozen at −80 °C, and tissue fixation was achieved by slowly thawing the organ samples in ice-cold formalin. After 4–7 days in formalin, tissue specimens were transferred to 70% ethanol, trimmed and routinely paraffin wax embedded. Consecutive sections (3–5 µm) were prepared and routinely stained with hematoxylin-eosin (HE) or subjected to immunohistology for the detection of PUUV N antigen in tissues.

Anti-PUUV N protein antiserum was generated by immunization (BioGenes GmbH, Berlin, Germany) of a single rabbit with PUUV N protein produced via baculovirus expression as described previously [27]. The same batch of PUUV N protein was used in an earlier diagnostic study [28]. Immunohistology was performed in an autostainer (Agilent) using the custom-made rabbit polyclonal antiserum and the horseradish peroxidase (HRP) method. Briefly, sections were deparaffinized and rehydrated through graded alcohol. Antigen retrieval was achieved by 20 min incubation in citrate buffer (pH 6.0) at 98 °C in a pressure cooker. This was followed by incubation with the primary antibody (diluted 1:1000 in dilution buffer; Dako) for 60 min at room temperature (RT), a 10 min incubation at room temperature (RT) with peroxidase blocking buffer (Agilent) and a 30 min incubation at RT with Envision+System HRP Rabbit (Agilent). The reaction was visualized with diaminobenzidin (DAB; Dako). After counterstaining with hematoxylin for 2 s, sections were dehydrated and placed on a coverslip with Tissue-Tek Film (Sysmex, Kobe, Japan). A formalin-fixed and paraffin embedded pellet of Vero E6 cells infected with PUUV for 14 days served as a positive control (infected cells exhibit a finely granular to coarse cytoplasmic staining).

### 2.11. PUUV-Specific Immunoglobulin Analysis

Immunofluorescence assays, using PUUV Sotkamo strain-infected Vero E6 cells fixed to microscope slides with acetone, were used to evaluate PUUV-specific immunoglobulin (Ig) in bank vole blood [29]. After incubating slides with blood diluted in PBS (1:10), bound Igs were detected with Fluorescein isothiocyanate (FITC)-conjugated rabbit anti-mouse Ig antibody (Agilent). Statistical differences between Ig titers in experimentally PUUV infected voles were assessed by one- or two-way ANOVA with Dunnett’s or Sivak’s multiple comparison tests as indicated in the figure legends. Analyses were conducted using GraphPad Prism version 8.1.2 (San Diego, CA, USA).

### 2.12. Ethics Statement

The Mygla.REC.B cell line was generated using an adult bank vole live trapped on the island of Riems, Greifswald, Germany. Permission for trapping was received from Landesamt für Landwirtschaft, Lebensmittelsicherheit und Fischerei Mecklenburg-Vorpommern 7221. 3-030/09. All vole trapping and experimental procedures performed in Finland were approved by the Animal Experimental Board of Finland (license number ESAVI/6935/04.10.07/2016).

## 3. Results

### 3.1. Isolation of PUUV-Suo in Epithelial Bank Vole Cells 

A bank vole renal epithelial cell line Mygla.REC.B was used to isolate a new PUUV strain named Suonenjoki (PUUV-Suo). Based on the detection of PUUV N protein by immunofluorescence (Figure 1), 100% of cells were infected after 20 cell passages post initial inoculation with the PUUV-containing lung homogenate. Further culturing of infected Mygla.REC.B cells up to 30 cell passages did not reduce the number of infected cells or the level of progeny viruses in cell culture supernatants (~10^4^ FFFU/mL; Figure 1B), indicating that Mygla.REC.B cells were persistently infected with PUUV-Suo.

The isolation process resulted in eight nucleotide exchanges when compared to the consensus sequence derived from the lung homogenate (Table 1). Two mutations occurred in non-coding regions, three resulted in amino acid exchanges in the viral polymerase (L segment), one resulted in an amino acid exchange in the N protein, and two were silent mutations. The new virus isolate had a 95%–99% nucleotide sequence identity with partially sequenced PUUV strains from bank voles in Konnevesi and from a fatal human case in Pieksämäki (Appendix A) [30,31], both located within 100 km of the vole trapping site in central Finland.

### 3.2. Replication of PUUV-Suo in Mygla.REC.B Cells Evades Antiviral Responses

We hypothesized that the observed persistent PUUV-Suo infection in bank vole cells would be a consequence of either impaired host cell antiviral signaling or the ability of the virus to counteract antiviral responses. Therefore, the antiviral function of Mygla.REC.B cells was assessed by relative quantification of the interferon-inducible Mx2 mRNA after exogenous activation of non-infected and PUUV-Suo infected cells with the TLR3 agonist polyI:C or Sendai virus. Firstly, supporting the immune competence of Mygla.REC.B cells, a significant upregulation of Mx2 mRNA occurred in non-infected cells with both stimulants (~100-fold and 300-fold by polyI:C and Sendai virus, respectively; Figure 2A). Secondly, the comparable Mx2 mRNA response in non-infected and infected cells suggested that replication of PUUV-Suo in Mygla.REC.B cells did not interfere with interferon signaling, at least in the context of polyI:C or Sendai virus treatment. Instead, as suggested by the similar levels of Mx2 mRNA between non-infected and PUUV-Suo infected cells (Figure 2B), the interferon-mediated immune system of Mygla.REC.B cells does not efficiently sense PUUV-Suo infection, which could facilitate a persistent PUUV-Suo infection in these cells.

### 3.3. Experimental PUUV-Suo Infection Resembles Natural Infection in Bank Voles

Following experimental infection of bank voles with PUUV-Suo or a mock inoculum (PBS), the amount of PUUV S segment RNA was measured in lungs, spleen, kidney and blood of voles at 3d, and 1, 2 and 5 wks pi. Viral RNA was measured in the urine of experimentally infected voles at 3d, and 1 and 2 wks pi. As another control, virus RNA load and RNA distribution in tissues were also assessed at 2 and 5 wks pi from voles inoculated with UV-inactivated PUUV-Suo isolate.

PUUV RNA was detected in all organs of PUUV-Suo infected voles from 3 dpi onwards (Figure 3A), but not in mock-infected animals nor those inoculated with UV-inactivated virus (data not shown). Highest PUUV RNA levels occurred in the lungs at 3d and 1 wk pi and decreased thereafter, with increased levels then observed in the spleen (at 2 and 5 wk pi) and kidneys (at 5 wk pi). PUUV RNA was detected in the blood of one vole at 3 dpi, suggesting that if PUUV-Suo caused viremia it occurred very early during infection. We also detected PUUV RNA in the urine of all tested PUUV-Suo infected bank voles (*n* = 7), suggesting virus shedding.

We further compared the infection dynamics of the PUUV-Suo isolate to a commonly used Vero E6 cell-adapted PUUV-Kazan isolate, a wild PUUV-containing lung homogenate (PUUV-wt) and to natural PUUV infections. The purified virus isolates PUUV-Suo and PUUV-Kazan used as inoculum displayed comparable viral protein and S segment RNA levels (Appendix A). Low levels of PUUV RNA were detected following infection with PUUV-Kazan, mainly in the lungs of infected voles at 3d pi; after this, the virus was efficiently cleared, with viral RNA levels significantly decreased at 1 wk pi (Figure 3B). No virus replication was observed for PUUV-wt at 2 wks pi, but three of four voles were positive at 5 wks pi (Figure 3C). At this time point, viral RNA loads were highest in the lungs, followed by the spleen and kidneys. The slow infection kinetics with PUUV-wt suggests low levels of infective virus in lungs of persistently infected bank vole, at least when compared to the PUUV-Suo isolate. The amount and distribution of PUUV RNA seen in naturally PUUV-infected wild bank voles (with unknown infection times) resembled that of PUUV-Suo infected voles during the first week and in voles inoculated with PUUV-wt at 5 wks pi (Figure 3D).

### 3.4. PUUV-Suo Host Cell Range Resembles Natural Infections 

Both naturally and experimentally infected bank voles were examined by histology and immunohistology for PUUV N protein. The two naturally infected voles did not exhibit any histopathological changes and viral antigen expression was limited in its amount and distribution. Besides endothelial cells in renal glomerula and in some capillaries of the liver, kidney, lungs and heart (Figure 4A,B), pneumocytes (mainly type II) and macrophages in liver (Kupffer cells) and spleen (red pulp macrophages) were found to be occasionally positive (Figure 4B). In one animal with a higher number of positive cells, viral antigen was also detected in tubular epithelial cells in the renal cortex and medulla.

None of the experimentally PUUV-Suo-infected voles exhibited histopathological changes. During the early stages of infection (≤1 wk pi), the infection pattern was similar to that seen in naturally infected bank voles. Capillary endothelial cells (also in glomerular tufts), pneumocytes and macrophages (Kupffer cells in the liver, pulmonary alveolar macrophages) occasionally exhibited viral antigen expression (Figure 4C,D). At later time points (2 and 5 wks pi), viral antigen was only found in macrophages, and almost exclusively in rare macrophages of the splenic red pulp (Figure 4E). No PUUV N protein was detected in PUUV-Kazan-infected voles, consistent with the low level of PUUV RNA in those animals (Figure 3C).

Histological examination of the PUUV-wt infected voles at 5 wks pi did not reveal any pathological changes. Interestingly, when testing lungs, kidneys and spleen by immunohistology, N protein expression was only detected in one of four animals, and only in the lungs, where a few individual pneumocytes and endothelial cells were positive (Figure 4F). This animal also showed the highest viral RNA levels in the lungs.

### 3.5. Delayed PUUV-Specific Immunoglobulin (Ig) Responses in Experimental Bank Vole Infections

All PUUV-Suo and PUUV-Kazan infected voles, which were infected for at least one week, seroconverted to produce PUUV-specific Ig but with contrasting kinetics (Figure 5). For PUUV-Suo, specific Ig-titers were low at 1 wk but peaked at 2 wks pi. This was significantly different from PUUV-Kazan infected voles, in which highest PUUV-specific Ig titers were detected at 1 wk pi. For PUUV-wt, only one of three infected voles had seroconverted at 5 wks pi (Figure 5) despite high viral loads in tissues (Figure 3D).

## 4. Discussion

In this study, we established and validated a new experimental orthohantavirus-reservoir host system. To achieve this, we isolated PUUV in cells from a bank vole host and observed minimal genetic changes when compared to wild PUUV. Analysis of Mx2 mRNA induction showed that the virus evaded antiviral responses during cell culture infection and persisted over cell passaging. Standardized experimental bank vole infections were then conducted, which demonstrated the ability of the new virus isolate to mimic wild-type infection properties in terms of viral load, tissue distribution, and host cell tropism. Furthermore, the new isolate shared the ability of wild PUUV to delay virus-specific humoral responses in experimentally infected voles. Taken together, our results indicate that orthohantavirus isolated in cells derived from its reservoir host retains wild-type infection characteristics and can be used as an experimental system to replicate natural processes.

The isolation of hantaviruses in cell culture is rare and has so far typically been achieved using interferon type 1-deficient cells (Vero E6 cells) derived from African green monkeys [32], which results in attenuated wild-type properties of the virus [12,13,14,15]. The isolation process of PUUV-Suo in bank vole renal epithelial cells resulted in a significantly lower nucleotide substitution frequency (0.06%) when compared to a PUUV-Kazan strain on Vero E6 cells [14,33] and cell culture infection that persisted cell passaging. This latter finding is consistent with the ability of wild PUUV to cause life-long infections in its reservoir host [34]. Despite the ability of Mygla.REC.B cells to respond to immune stimulus by inducing Mx2 mRNA expression, cells infected with PUUV-Suo did not display significantly altered Mx2 mRNA levels, which suggested that PUUV-Suo is able to evade stimulation of interferon-mediated innate immune pathways.

The PUUV-Suo isolate caused persistent vole infections without pathological effects. Target cell patterns and viral RNA loads, particularly during the early stages of infection, were consistent with those seen in naturally infected voles and were considerably different from experimental infections conducted with the attenuated PUUV-Kazan isolate. The viral target cell pattern observed for PUUV-Suo is similar to, but less widespread than a previously described experiment in which weanling and suckling bank voles were inoculated with lung homogenates from PUUV-positive bank voles [11]. Experimental infections conducted with Sin Nombre orthohantavirus (SNV) in deer mice yielded comparable, though apparently more intense viral antigen expression [10]. In naturally infected deer mice, however, only pulmonary and cardiac vascular endothelial cells were found to be positive, with an expression intensity that resembled that observed in the present study [35]. Interestingly, only PUUV-Suo appears to infect renal tubular epithelial cells. This could reflect different primary transmission routes for PUUV and SNV, with PUUV suggested to be shed more in urine and SNV in the saliva of natural reservoir systems [6].

Our finding that PUUV-specific Ig responses were significantly delayed in voles infected with PUUV-Suo was unexpected, as was the lack of seroconversion in two of three bank voles infected with PUUV-wt. This is due to previous observations of strong virus-specific IgG responses and high neutralizing antibody titers in wild hantavirus infected rodents [36,37,38]. However, comparisons between wild-type and attenuated hantavirus phenotypes have not been previously performed in animal models. Our results suggest that wild PUUV has the ability to delay immune recognition in bank voles early during infection, which could enable the virus to spread among tissues in the host. Once a persistent infection has been established, even strong neutralizing humoral responses may then be inadequate to clear the virus. While the molecular mechanism behind the observed delay in PUUV-specific responses by PUUV-Suo is unclear, it is consistent with the ability of PUUV-Suo to evade antiviral responses in bank vole cells in vitro (Figure 2). It is likely that the attenuated PUUV-Kazan strain has lost its ability to evade this recognition during the adaptation process to the interferon-deficient Vero E6 cell line.

Our results show that a hantavirus isolated using allogeneic reservoir host cells retains wild-type characteristics better than a Vero E6-adapted virus and should be considered as the preferred method for future hantavirus isolations and experimental infections to mimic natural processes. The establishment of such animal models will facilitate future studies on the detailed molecular mechanisms of persistence in hosts, natural patterns of hantavirus transmission and human disease models.

## Figures and Tables

**Figure 1 viruses-12-00457-f001:**
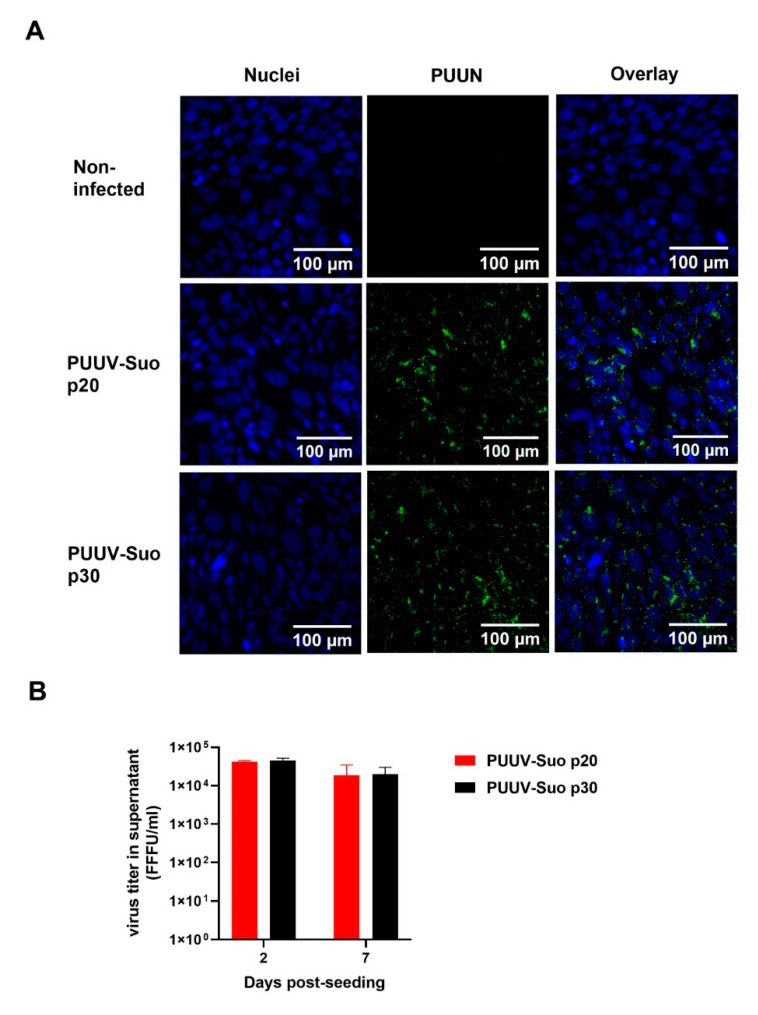
Successful isolation of PUUV-Suo in a bank vole cell line. (**A**) Non-infected or PUUV-Suo infected bank vole kidney epithelial cells (Mygla.REC.B) were stained for nuclei (DNA) using Hoechst 33420 (blue) and for PUUV nucleocapsid protein (PUUN) using PUUN-specific rabbit polyclonal antibody followed by AlexaFluor488-conjugated secondary antibody (green) followed by confocal microscopy analysis. PUUV-Suo infected cells at cell passages 20 or 30 post exposure to wild PUUV containing lung homogenates are shown. Images are obtained from cells grown for 2 days post-seeding. (**B**) Virus titers were measured in supernatants of PUUV-Suo infected Mygla.REC.B cells (passages 20 and 30) at 2 and 7 days post-seeding. Bars represent mean +/− standard deviation (*n* = 2).

**Figure 2 viruses-12-00457-f002:**
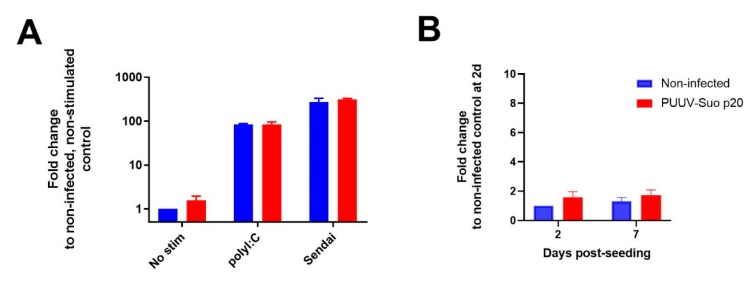
Effect of PUUV-Suo infection on interferon-stimulated Mx2 mRNA expression in Mygla.REC.B cells. (**A**) Non-infected or PUUV-Suo infected Mygla.REC.B cells (passage 20 post inoculation with wild PUUV) were either non-stimulated, stimulated with polyI:C or infected with Sendai virus for 24 h after 2 days post-seeding. RNA was isolated from cells and subjected to relative quantification of Mx2 mRNA using RT-qPCR using actin mRNA levels as reference. (**B**) Relative quantification of Mx2 mRNA in non-infected and PUUV-Suo infected Mygla.REC.B cells was performed from cells grown for 2 and 7 days post-seeding. Bars represent mean +/− standard deviation (*n* = 2).

**Figure 3 viruses-12-00457-f003:**
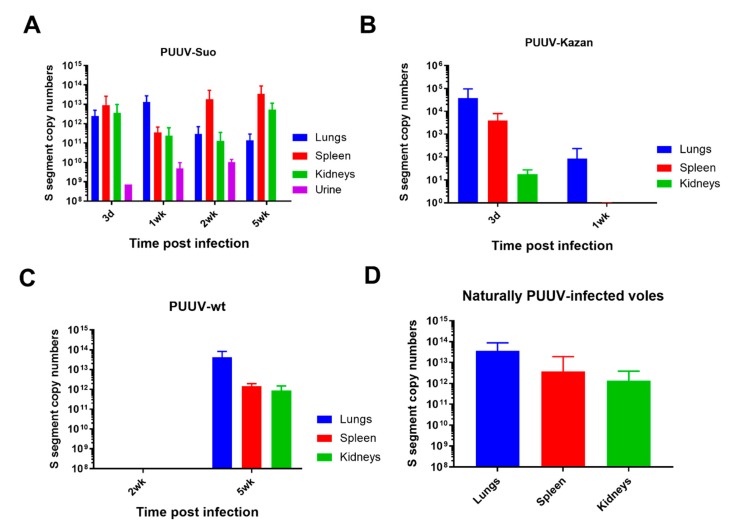
Viral RNA load analysis of experimentally and naturally infected bank voles. The lungs, spleen and kidneys of bank voles infected with PUUV-Suo (**A**, *n* = 4), PUUV-Kazan (**B**, *n* = 3) or PUUV-wt (**C**, *n* = 3) were collected at indicated times post infection to quantify PUUV copy numbers by PUUV S segment-specific qPCR. In the case of PUUV-Suo, urine was also included in the analysis (*n* = 0–4 per time point). In addition, seropositive naturally PUUV-infected bank voles with unknown infection times were included as reference (**D**, *n* = 27–38). Bars represent mean + standard deviation. Results show comparable viral loads in voles infected with PUUV-Suo and PUUV-wt as compared to natural infections.

**Figure 4 viruses-12-00457-f004:**
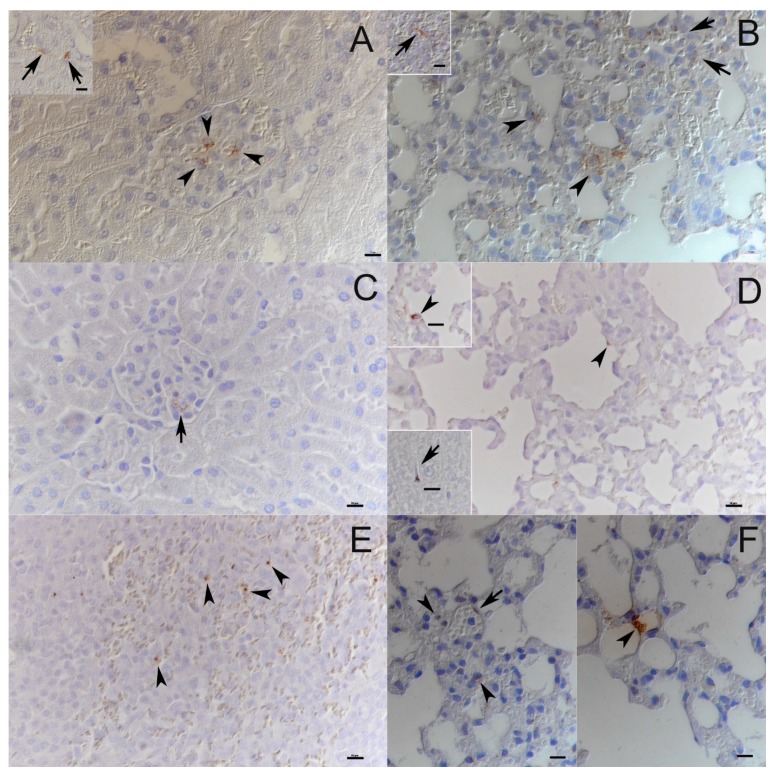
PUUV target cell analysis in experimental and natural infections. (**A**) Kidneys of naturally infected bank voles. Viral antigen expression is apparent in occasional glomerular endothelial cells (arrowheads) and occasional vascular endothelial cells (inset: arrows). (**B**) Lungs and spleen of naturally infected bank voles. A variable number of vascular/capillary endothelial cells (arrows) and pneumocytes (arrowheads) express viral antigen in the lungs. Very few cells in the splenic red pulp, consistent with macrophages, are also found to express viral antigen (inset: arrow). (**C**) Kidney of PUUV-Suo infected bank vole at 1 wk pi show scattered glomerular endothelial cells with viral antigen expression. (**D**) The lungs of PUUV-Suo-infected voles at 3 d pi exhibit occasional positive alveolar epithelial cells (arrowhead, also in top inset), and scattered positive Kupffer cells in the liver (bottom inset: arrow). (**E**) The spleen of PUUV-Suo-infected voles at 2 wks pi show very limited viral antigen expression, which is restricted to individual macrophages, for example, in the splenic red pulp (arrowheads). (**F**) The lungs of PUUV-wt-infected bank voles at 5 wks pi show viral antigen expression restricted to occasional vascular endothelial cells (left, arrow) and scattered pneumocytes in the lungs (right, arrowhead). The images are representative of two voles analyzed by immunohistochemistry for PUUV N protein expression in naturally infected voles and at each time point of PUUV-Suo or PUUV-wt infections. Haematoxylin counterstain. Bars = 10 µm.

**Figure 5 viruses-12-00457-f005:**
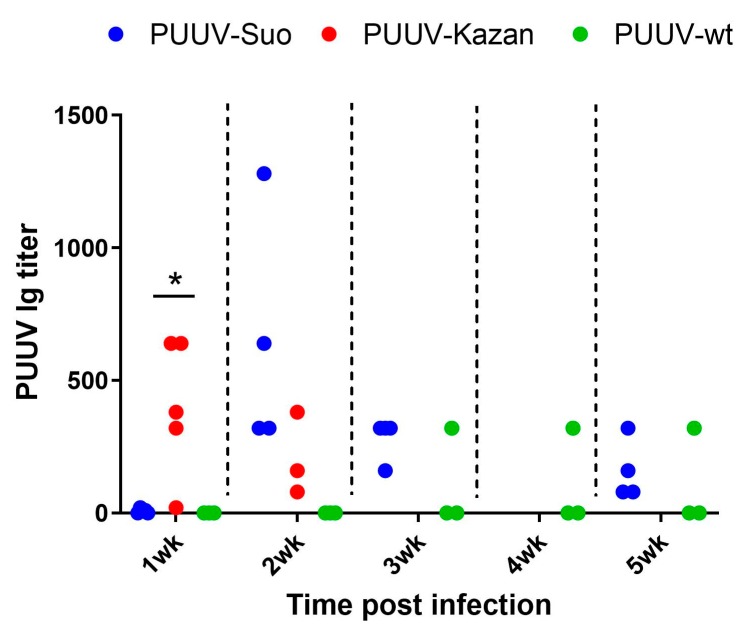
Assessment of PUUV-specific Ig responses in blood of experimentally and naturally infected bank voles. PUUV-specific Ig titers in blood of PUUV-Suo (*n* = 4), PUUV–Kazan (*n* = 3–5) and PUUV–wt (*n* = 3) infected voles at indicated time points. * indicates statistically significant difference (*p* < 0.05).

**Table 1 viruses-12-00457-t001:** Comparison between the consensus nucleotide sequences of the PUUV-Suo strain at the pre-isolation stage in the lungs of a naturally infected bank vole and after isolation in Mygla.REC.B cells.

Segment	Nucleotide Position in Anti-Genomic Orientation	Nucleotide in Original Virus	Nucleotide in Isolate	Amino Acid in Original Virus	Amino Acid in Isolate	Codon Position
L	66	A	G	no change		3rd
	411	G	A	Methionine	Isoleucine	3rd
	5700	A	G	Isoleucine	Methionine	3rd
	5942	C	U	Alanine	Valine	2nd
M	89	U	C	no change		1st
	3544	U	C			non-coding
S	136	G	U	Valine	Leucine	1st
	1625	C	U			non-coding

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
