# Peer review of "Orthohantavirus Isolated in Reservoir Host Cells Displays Minimal Genetic Changes and Retains Wild-Type Infection Properties"

_viruses, 2020, doi:10.3390/v12040457_

Round 1

Reviewer 1 Report

The authors report the isolation and characterization of a novel Puumala orthohantavirus. Isolation of hantaviruses is challenging and is usually done in VeroE6 cells, which unfortunately leads to cell culture adaptation and even attenuation in animal models of disease. In this study, the authors used immortalized cells derived from its reservoir host for the isolation of Puumala virus. This resulted in persistent cell culture infection, efficient immune evasion and little viral genome sequence changes when compared to the original virus. Most interesting, experimental infections of bank voles with the new isolate resembled natural infections more closely. The experiments were well planned and implemented.

The results of this study are very interesting, as they present a possibility to limit viral adaptation or attenuation during the propagation process. Therefore, the usage of viruses isolated and propagated in reservoir host cells in animal models may be an important alternative to the difficult inoculation of virus obtained directly from reservoir hosts.

Minor spellcheck is required throughout. 

Minor comments:

  • Did the authors try to infect the Mygla.REC.B cells with PUUV Kazan? Does this lead to a productive (persistent) infection? It would be interesting to compare innate immune induction by PUUV-Kazan and PUUV-Suo.
  • It would be interesting to see if PUUV-Suo looses its "WT-like" abilities after passage on Vero E6 cells.
  • page 1, line 100-102: "Both PUUV-Suo and PUUV-Kazan strains were purified from supernatants of infected cells through a 30 % sucrose cushion by ultracentrifugation and resuspended in corresponding growth medium." Why were different growth media used? In terms of comparability it would have been beneficial to dissolve both viruses in the same media after purification.
  • page 3, line 109: Which passage of PUUV-Suo was used for sequencing?
  • page 9, figure 3C: The late increase in viral copy numbers in PUUV-wt infected voles may be explained by lower virus-concentrations in the inocculum? The different kinetics should be discussed.
  • page 12, line 392 ff: It would be interesting to see if PUUV-Suo infection leads to DC activation; or differences in PUUV-Kazan or PUUV-Suo mediated MHC I antigen presentation in infected endothelial cells.

minor corrections:

page 1, line 41-44: The sentence could be rearranged to improve readability

Author Response

Reviewer1:

The authors report the isolation and characterization of a novel Puumala orthohantavirus. Isolation of hantaviruses is challenging and is usually done in VeroE6 cells, which unfortunately leads to cell culture adaptation and even attenuation in animal models of disease. In this study, the authors used immortalized cells derived from its reservoir host for the isolation of Puumala virus. This resulted in persistent cell culture infection, efficient immune evasion and little viral genome sequence changes when compared to the original virus. Most interesting, experimental infections of bank voles with the new isolate resembled natural infections more closely. The experiments were well planned and implemented.

The results of this study are very interesting, as they present a possibility to limit viral adaptation or attenuation during the propagation process. Therefore, the usage of viruses isolated and propagated in reservoir host cells in animal models may be an important alternative to the difficult inoculation of virus obtained directly from reservoir hosts.

Minor spellcheck is required throughout.

The manuscript has been checked and several minor grammatical changes made.  

Minor comments:

Did the authors try to infect the Mygla.REC.B cells with PUUV Kazan? Does this lead to a productive (persistent) infection? It would be interesting to compare innate immune induction by PUUV-Kazan and PUUV-Suo.

We have been unable to reach similar infection levels with PUUV-Kazan to those observed with PUUV-Suo and conclude that MyGla.REC.B cells are not highly susceptible to PUUV-Kazan infection. This precludes proper comparisons between the two virus isolates in terms of persistence or innate immunity activation. 

It would be interesting to see if PUUV-Suo looses its "WT-like" abilities after passage on Vero E6 cells.

This is a good point and we are in the process of passaging PUUV-Suo in Vero E6 cells. Unfortunately, for the current study, we don’t have the possibility to investigate the infectivity of this virus in wild bank voles further. This is due to a lack of wild voles consistent with their seasonal and multi-annual density fluctuations as well as the animal facility being closed by the coronavirus pandemic.

page 1, line 100-102: "Both PUUV-Suo and PUUV-Kazan strains were purified from supernatants of infected cells through a 30 % sucrose cushion by ultracentrifugation and resuspended in corresponding growth medium." Why were different growth media used? In terms of comparability it would have been beneficial to dissolve both viruses in the same media after purification.

MyGla.Rec.B  and Vero E6 cells require different growth mediums and we used the same medium as used for infected-cell cultivation for resuspending each of the purified viruses. While this may not have been optimal for comparisons between viruses, we later suspended MyGla.Rec.B-produced PUUV-Suo in Vero E6 growth medium without observing differential effects on infectivity in cell culture or bank vole infections.

page 3, line 109: Which passage of PUUV-Suo was used for sequencing?

Passage 20 was used. This has now been added to the manuscript text.

page 9, figure 3C: The late increase in viral copy numbers in PUUV-wt infected voles may be explained by lower virus-concentrations in the inocculum? The different kinetics should be discussed.

This is true. We have added discussion on this: “The slow infection kinetics with PUUV-wt suggests low levels of infective virus in lungs of persistently infected bank vole, at least when compared to the PUUV-Suo isolate.” page 8, line 296.

page 12, line 392 ff: It would be interesting to see if PUUV-Suo infection leads to DC activation; or differences in PUUV-Kazan or PUUV-Suo mediated MHC I antigen presentation in infected endothelial cells.

We agree. These suggestions are interesting and hopefully are something that we could look into in the future. Currently, the lack of specific antibodies or other methods for identifying different cell types in bank voles hinders us from more detailed studies on the mechanisms of hantavirus persistence.

minor corrections:

page 1, line 41-44: The sentence could be rearranged to improve readability

We changed curtail to combat. It now reads: “Understanding how zoonotic pathogens are maintained and transmitted in nature is critical for efforts to combat human disease”.

Reviewer 2 Report

Strandin, et al. present the characterization of Puumala virus isolated directly from a bank vole and compare the differences of this virus when passaged through a bank vole primary cell line to Puumala virus passaged through Vero E6 cells and wild-type Puumala virus. This publication provides an interesting approach to compare the effects of passaging on antiviral responses and guides future modeling efforts. This manuscript is well-written, and comments are minor:

  1. Fig 3. Was urine not collected in PUUV-Kazan, PUUV-wt and naturally infected voles?
  2. Fig 5. Is there not a PUUV-Suo 4 week timepoint? Similarly, PUUV-Kazan 3, 4 and 5 wk timepoint?
  3. The authors’ state that voles infected with PUUV-wt had high viral loads noted in Fig 3C. However, only 1/3 animals seroconverted in contrast to cited references indicating that a robust Ig response was anticipated. Perhaps an n>3 might be necessary.

Author Response

Reviewer2:

Strandin, et al. present the characterization of Puumala virus isolated directly from a bank vole and compare the differences of this virus when passaged through a bank vole primary cell line to Puumala virus passaged through Vero E6 cells and wild-type Puumala virus. This publication provides an interesting approach to compare the effects of passaging on antiviral responses and guides future modeling efforts. This manuscript is well-written, and comments are minor:

    Fig 3. Was urine not collected in PUUV-Kazan, PUUV-wt and naturally infected voles?

Unfortunately, we collected urine only from PUUV-Suo infected voles. Based on previous publications (Lundkvist et al., 1997 and Klingstrom et al., 2002), we did not expect the PUUV-Kazan shedding from bank voles and elected not to collect urine samples to limit  animal stress (keeping animals alone in metabolic cages). For PUUV-wt voles, in hindsight, urine collection would have been a good way of detecting productive infection without the need to sacrifice animals. However, after the first set of experiments, we felt the added value of urine collection was not enough to justify another infection experiment with PUUV-wt. Finally, naturally infected voles were sacrificed, and organs collected for PCR, directly after capture. These animals were not brought back to the lab and therefore systematic urine collection was not possible. 

    Fig 5. Is there not a PUUV-Suo 4 week timepoint? Similarly, PUUV-Kazan 3, 4 and 5 wk timepoint?

That is correct. Only the PUUV-wt infected voles were subjected to blood sampling at the 4-week time point.  In the case of PUUV-Suo infected voles, all animals had already seroconverted at this point, and considering animal welfare, the added value of additional sampling was minimal. Due to same reasons, we elected not to continue sampling PUUV-Kazan infected voles past the 2-week time point.   

    The authors’ state that voles infected with PUUV-wt had high viral loads noted in Fig 3C. However, only 1/3 animals seroconverted in contrast to cited references indicating that a robust Ig response was anticipated. Perhaps an n>3 might be necessary.

The cited references indicate that experimentally hantavirus-infected reservoir hosts typically show high Ig titer seroconversion at certain times post-inoculation. However, no comparison between wild-type and attenuated hantaviruses has been made previously. We have added the following sentence to the discussion section of the manuscript to make this point clear: “ However, comparisons between wild-type and attenuated hantavirus phenotypes have not been previously performed in animal models” page 12, lines 398-399.

We agree that a higher N would improve the robustness of our finding of slow seroconversion regarding PUUV-wt. However, we feel results with PUUV-wt support our main finding of slow seroconversion observed in PUUV-Suo infected voles when compared to the attenuated PUUV-Kazan virus.